# Increase in clinically recorded type 2 diabetes after colectomy

Anders B Jensen[1,2†], Thorkild IA Sørensen[3,4†], Oluf Pedersen[3], Tine Jess[5], Søren Brunak[1‡*], Kristine H Allin[3,5‡*]

[1]Novo Nordisk Foundation Center for Protein Research, Faculty of Health and Medical Sciences, University of Copenhagen, Copenhagen, Denmark; [2]Institute for Next Generation Healthcare, Mount Sinai Health System, New York, United States; [3]Novo Nordisk Foundation Center for Basic Metabolic Research, Section of Metabolic Genetics, Faculty of Health and Medical Sciences, University of Copenhagen, Copenhagen, Denmark; [4]Department of Public Health, Section of Epidemiology, Faculty of Health and Medical Sciences, University of Copenhagen, Copenhagen, Denmark; [5]Center for Clinical Research and Prevention, Bispebjerg and Frederiksberg Hospital, Copenhagen, Denmark

*For correspondence:
soren.brunak@cpr.ku.dk (SøB);
kristine.allin@regionh.dk (KHA)

†These authors contributed equally to this work
‡These authors also contributed equally to this work

**Competing interests:** The authors declare that no competing interests exist.

**Abstract** The colon hosts gut microbes and glucagon-like peptide 1 secreting cells, both of which influence glucose homeostasis. We tested whether colectomy is associated with development of type 2 diabetes. Using nationwide register data, we identified patients who had undergone total colectomy, partial colectomy, or proctectomy. For each colectomy patient, we selected 15 non-colectomy patients who had undergone other surgeries. Compared with non-colectomy patients, patients with total colectomy (n = 3,793) had a hazard ratio (HR) of clinically recorded type 2 diabetes of 1.40 (95% confidence interval [CI], 1.21 to 1.62; p<0.001). Corresponding HRs after right hemicolectomy (n = 10,989), left hemicolectomy (n = 2,513), and sigmoidectomy (n = 13,927) were 1.08 (95% CI, 0.99 to 1.19; p=0.10), 1.41 (95% CI, 1.19 to 1.67; p<0.001) and 1.30 (95% CI, 1.21 to 1.40; p<0.001), respectively. Although we were not able to adjust for several potential confounders, our findings suggest that the left colon may contribute to maintenance of glucose homeostasis.
DOI: https://doi.org/10.7554/eLife.37420.001

## Introduction

The gut acts as a key regulator of glucose homeostasis. It influences appetite, gastric emptying, intestinal motility, digestion, and absorption of nutrients, as well as insulin secretion from the pancreas through hormonal secretion and nervous signaling (*Holst et al., 2016*). Whereas the role of the small intestine in glucose homeostasis has been well elucidated (*Holst and Madsbad, 2016*), the role of the colon is less clear. The growing interest in the gut microbiota has led to an increased focus on colonic pathophysiology since the vast majority of gut microbes resides in the colon. It has been shown that patients with type 2 diabetes harbor an altered fecal microbiota (*MetaHIT consortium et al., 2015*; *Karlsson et al., 2013*; *Qin et al., 2012*), but it is unknown whether it is causally involved in the pathogenesis of type 2 diabetes (*Allin et al., 2015*; *IMI-DIRECT consortium et al., 2018*; *MetaHIT Consortium et al., 2016*). Whereas some mechanistic studies in mice suggest that obesity and its related phenotypes can be transmitted via the fecal microbiota (*Bäckhed et al., 2004*; *Ridaura et al., 2013*), elimination of the gut microbiota by antibiotics in humans, does not seem to have any short-term effect on glucose regulation (*Mikkelsen et al., 2015*; *Reijnders et al., 2016*; *Vrieze et al., 2014*).

**eLife digest** The large intestine helps people process the food that has not been digested and absorbed in the small intestine. It houses billions of bacteria that break down food, especially fibers. It also produces hormones that may help control blood sugar after eating. People with type 2 diabetes, who have difficulty controlling their blood sugar after meals, have a different mix of gut bacteria than people without the disease. But scientists do not know if this unusual mix of bacteria is a result of having diabetes or its treatment, or if it contributes to the condition. They also do not know if hormones released by the large intestine play a role in the disease.

One way to study the role of the large intestine in diabetes is to look at patients who have had all or part of it removed. A few small studies with 21 patients or fewer have looked at how well patients are able to control their blood sugar after removal of all or part of their large intestine. But the results were confusing.

Now, Jensen, Sørensen et al. show that patients who have surgery to remove all or the left part of their large intestine are more likely to develop diabetes. They looked at patient records from a national registry in Denmark to see how many patients developed diabetes during up to 18 years after surgery. In the analysis, they compared post-surgical diabetes diagnoses in 3,793 people who had their whole large intestine removed, 42,486 who had part of it removed, and 694,110 people who had surgery on another part of the body. People who had their rectum removed did not have a greater risk of developing diabetes than people having other surgeries. But the risk was higher among those who had all or the left part of the large intestine taken out. This suggests that the left part of the large intestine helps people control their blood sugar levels.

More studies are needed to confirm that the large intestine plays a role in diabetes and to identify the underlying mechanisms. Understanding how the large intestine helps control blood sugar may help scientists develop new ways of treating or preventing type 2 diabetes.

DOI: https://doi.org/10.7554/eLife.37420.002

In addition to hosting the gut microbiota, the colon also serves as an endocrine organ. Glucagon-like peptide 1 (GLP-1) secreted by L-cells in the small intestine augments insulin secretion from the pancreatic β-cells in response to a meal (*Holst et al., 2016*). L-cells are, however, not only present in the small intestine but also in the colon, where the cell density increases from the proximal colon to the rectum (*Eissele et al., 1992*; *Gunawardene et al., 2011*; *Jorsal et al., 2018*). Accordingly, it has been shown that biologically active GLP-1 is present in colonic tissue in concentrations, which are similar to those in the small intestine (*Deacon et al., 1995*), but the physiological role of colonic GLP-1 remains elusive. In addition to GLP-1, colonic L cells also secrete the appetite-reducing hormone peptide YY (PYY) (*Svane et al., 2016*).

Patients who have undergone resection of the entire colon or parts of it, here denoted colectomy, may serve as an evident opportunity to study the role of the colon in health and disease (*Jensen et al., 2015*). Glucose regulation in patients with colectomy has been examined in a few previous studies, the largest of which included 21 patients, and although results are not clear-cut, they suggest that colectomy may influence the entero-insulinar axis (*Besterman et al., 1982*; *Nauck et al., 1996*; *Palnaes Hansen et al., 1997*). On one hand, based on studies showing that mice raised in the absence of microorganisms have less body fat and are less insulin resistant, removal of the human gut microbiota by colectomy may be expected to result in a reduced risk of type 2 diabetes (*Bäckhed et al., 2004*). On the other hand, removal of GLP-1 and PYY producing L-cells by colectomy may be expected to lead to an increased risk of type 2 diabetes.

In the present study, we tested the hypothesis that total and partial colectomy is associated with an altered risk of type 2 diabetes. For this purpose, we used the nationwide Danish National Patient Register to compare the frequency of clinically recorded type 2 diabetes in patients who had a colectomy with patients who had other types of surgeries not involving the gastrointestinal tract.

## Results

We identified 8946 individuals with total colectomy and 90,594 individuals with partial colectomy or proctectomy in the Danish National Patient Register. As explained in detail in Materials and

methods, we based our main analyses on those who were alive and who did not have a diagnosis of diabetes recorded during the first 1000 days after the surgery. Patients with records of total colectomy at different dates or more than one partial colectomy, patients younger than 30 years, patients who died within the first 1000 days of follow-up, patients diagnosed with diabetes before surgery or within the first 1000 days after surgery, and patients who had a follow-up time shorter than 1000 days were excluded (*Figure 1*). Thus, we included 46,279 patients who had undergone total colectomy, partial colectomy, or proctectomy. For comparison, we identified a total of 694,110 age, sex, and year of surgery matched non-colectomy patients who had undergone other types of surgery not involving the gastrointestinal tract (*Table 1*). Mean (SD) follow-up time was 7.1 (5.2), 5.6 (4.6), 6.3 (4.7), 5.7 (4.8), 6.4 (4.9), 5.8 (4.8) years for patients with total colectomy, right hemicolectomy, resection of the transverse colon, left hemicolectomy, sigmoidectomy, and proctectomy, respectively.

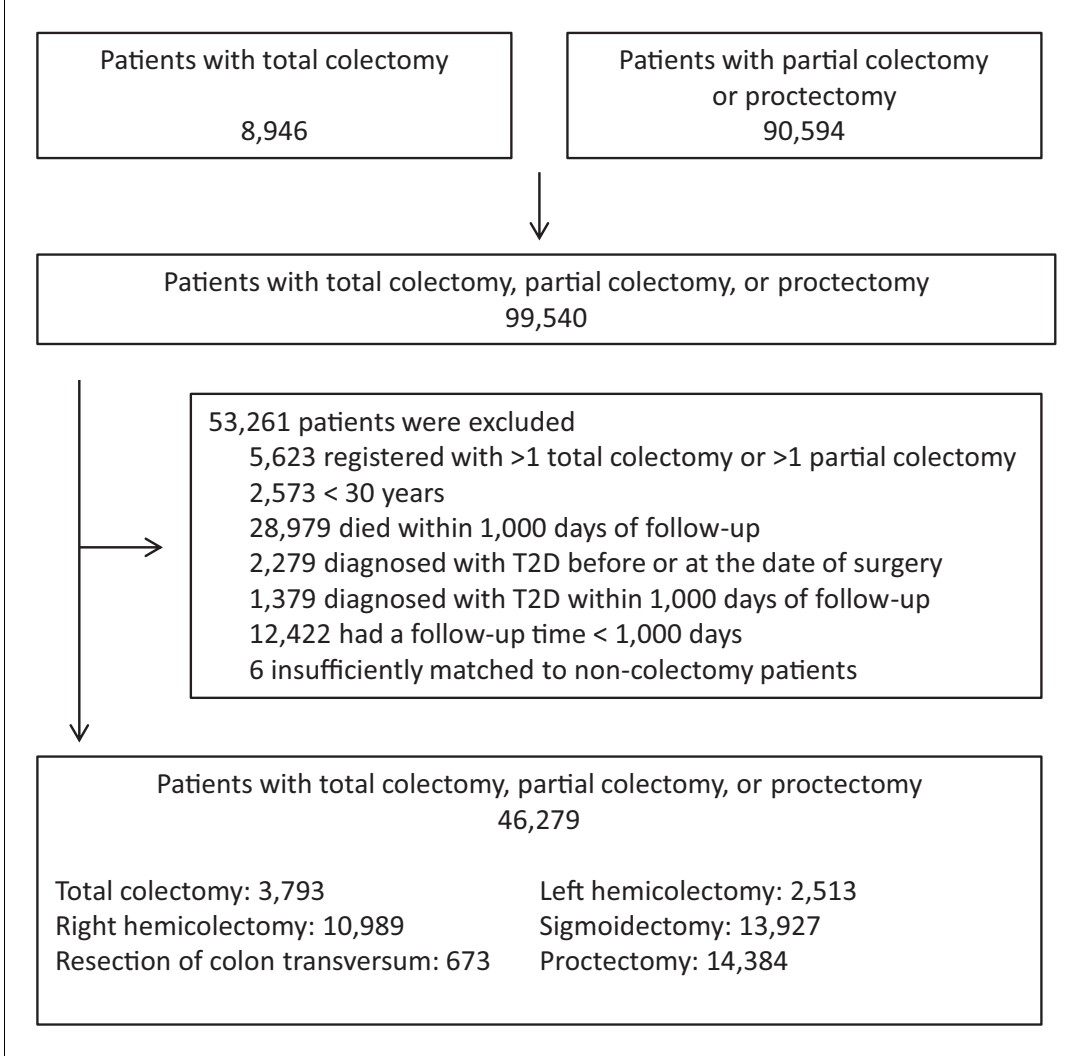

**Figure 1.** Study design. The chart illustrates the number of eligible patients, number of patients excluded and reasons for exclusions, and the final number of patients included. For each colectomy patient, 15 non-colectomy patients were selected among (1) patients who had undergone orthopedic surgery, (2) patients who had undergone abdominal surgery leaving the gastrointestinal tract intact, and (3) patients who had undergone other surgery, unrelated to the gastrointestinal tract. Five non-colectomy patients were selected from each surgery group resulting in inclusion of a total of 694,110 non-colectomy patients matched on age, sex, and year of surgery. Non-colectomy patients were selected using sampling with replacement wherefore the total number of matches are slightly lower than the total number of colectomy patients times 15. Total colectomy includes colectomy and proctocolectomy. T2D: clinically recorded type 2 diabetes.

DOI: https://doi.org/10.7554/eLife.37420.003

**Table 1.** Characteristics of the study population.

| | Patients with colectomy or proctectomy | | Non-colectomy patients | | | | | | | | |
| | | | All | | Orthopedic surgery | | Abdominal surgery leaving the GI tract intact | | Other surgery unrelated to the GI tract | |
| | No. (% women) | Age, years | No. (% women) | Age, years | No. (% women) | Age, years | No. (% women) | Age, years | No. (% women) | Age, years |
|---|---|---|---|---|---|---|---|---|---|---|
| Total colectomy | 3793 (50) | 54.6 (15) | 56,895 (50) | 54.6 (15) | 18,965 (50) | 54.6 (15) | 18,965 (50) | 54.6 (15) | 18,965 (50) | 54.6 (15) |
| Right hemicolectomy | 10,989 (61) | 68.1 (13) | 164,815 (60) | 68.0 (13) | 54,940 (60) | 68.0 (13) | 54,940 (60) | 68.0 (13) | 54,935 (60) | 67.9 (13) |
| Resection of colon transversum | 673 (61) | 66.6 (14) | 10,085 (60) | 66.5 (14) | 3365 (61) | 66.5 (14) | 3360 (60) | 66.5 (13) | 3360 (60) | 66.5 (14) |
| Left hemicolectomy | 2513 (53) | 66.4 (12) | 37,685 (53) | 66.3 (12) | 12,560 (53) | 66.3 (12) | 12,560 (53) | 66.3 (12) | 12,565 (53) | 66.2 (12) |
| Sigmoidectomy | 13,927 (53) | 65.0 (13) | 208,880 (53) | 64.9 (13) | 69,635 (53) | 65.0 (13) | 69,620 (53) | 64.9 (13) | 69,625 (53) | 64.9 (13) |
| Proctectomy | 14,384 (48) | 65.7 (12) | 215,750 (48) | 65.6 (12) | 71,920 (48) | 65.6 (12) | 71,915 (48) | 65.6 (12) | 71,915 (48) | 65.6 (12) |

Total colectomy includes colectomy and proctocolectomy. Age is mean (SD). GI: Gastrointestinal tract

DOI: https://doi.org/10.7554/eLife.37420.004

The outcome was routinely clinically diagnosed type 2 diabetes (hereafter denoted 'diabetes') as recorded in the Danish National Patient Register. *Figure 2* shows the time-corresponding cumulative hazards (double Nelson-Aalen plot) of diabetes for colectomy patients vs. non-colectomy patients. The slopes of the curves equalize the hazard ratios (a slope of 1.00 corresponds to a HR of 1.00). The cumulative hazard of diabetes was greater among patients who had undergone total colectomy compared with non-colectomy patients. Correspondingly, the HR of diabetes was 1.40 (95% confidence interval [CI], 1.21 to 1.62; p<0.001) (*Figure 3*). The analysis of the risk of diabetes after partial colectomy revealed that the highest cumulative hazard was observed among patients who had undergone left hemicolectomy or sigmoidectomy. Accordingly, the HR was 1.41 (95% CI, 1.19 to 1.67; p<0.001) for patients who had undergone left hemicolectomy and 1.30 (95% CI, 1.21 to 1.40; p<0.001) for patients who had undergone sigmoidectomy. Right hemicolectomy was not statistically significantly associated with increased risk of diabetes (HR 1.08 [95% CI, 0.99 to 1.19; p=0.10]). Resection of the transverse colon was infrequent, but showed the same estimate, although with wider confidence interval; HR 1.08 (95% CI, 0.76 to 1.54; p=0.66), whereas proctectomy was not associated with increased risk of diabetes; HR 0.98 (95% CI, 0.91 to 1.07; p=0.71).

We stratified patients with colectomy into a group of colorectal cancer patients and a group with other colorectal diseases (*Supplementary file 1* lists the most common diagnoses for the patients with other colorectal diseases). In both groups of patients, total colectomy was associated with increased risk of diabetes (*Table 2*). The HR of diabetes after total colectomy associated with colorectal cancer was 1.61 (95% CI, 1.22 to 2.11, p<0.001), whereas the HR for total colectomy associated with other colorectal disease was 1.34 (95% CI, 1.13 to 1.59, p<0.001). Corresponding HRs were 1.24 (95% CI, 1.01 to 1.53, p=0.04) and 1.88 (95% CI, 1.41 to 2.50, p<0.001) for left hemicolectomy, and 1.11 (95% CI, 0.997 to 1.24, p=0.06) and 1.47 (95% CI, 1.34 to 1.62, p<0.001) for sigmoidectomy.

If we, instead of following the patients from 1000 days after surgery, followed the patients from the date of surgery, we found a somewhat higher HR of diabetes after total colectomy: HR = 1.72 (95% CI, 1.56 to 1.89), which, however, should be interpreted with caution due to non-proportional hazards. In contrast, HRs of diabetes after left hemicolectomy and sigmoidectomy remained similar to results from the main analyses (*Supplementary file 2*). When follow-up time began 500 days after surgery or 1500 days after surgery we observed similar HRs of diabetes as compared to the main analyses (*Supplementary file 2*).

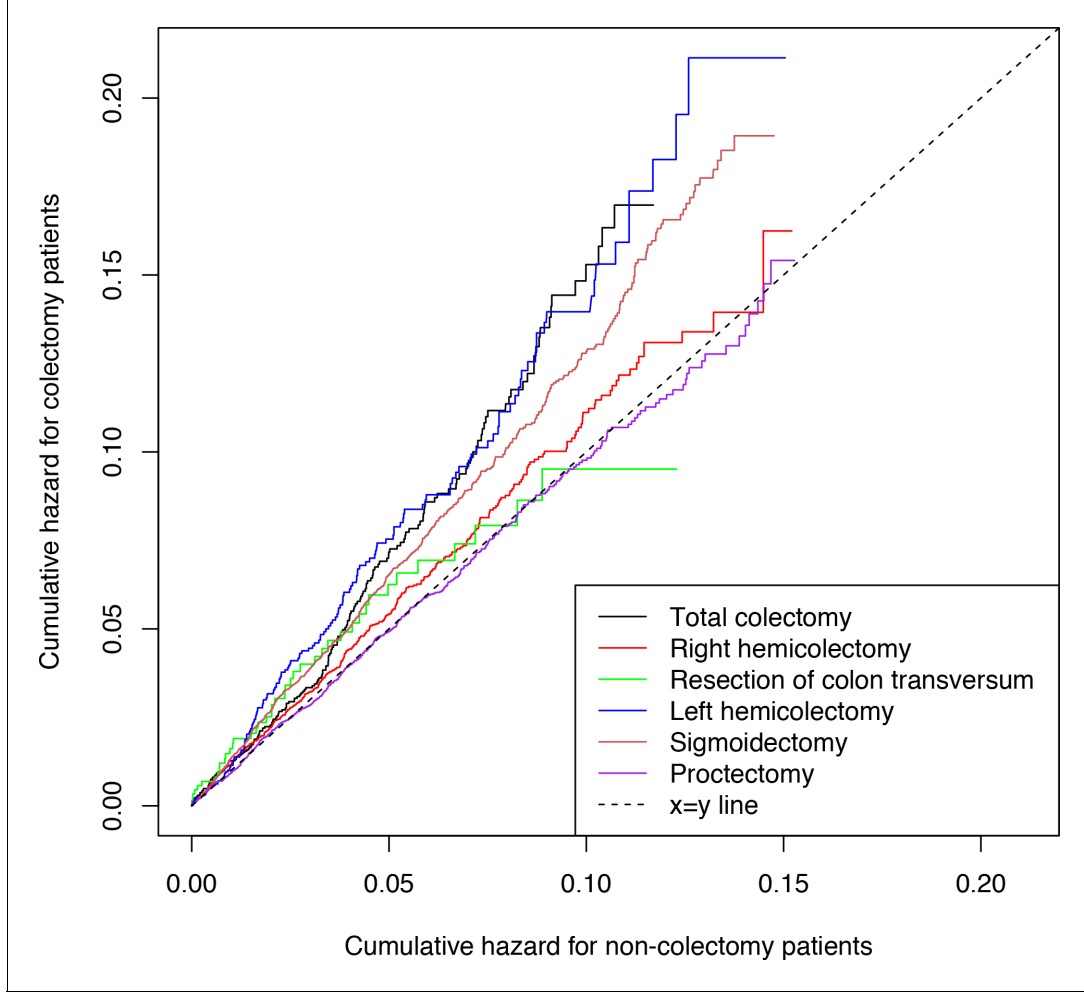

**Figure 2.** Cumulative hazards of clinically recorded type 2 diabetes. Cumulative hazards are presented for patients with total colectomy, right hemicolectomy, resection of colon transversum, left hemicolectomy, sigmoidectomy, and proctectomy and for non-colectomy patients. Total colectomy includes colectomy and proctocolectomy. The cumulative hazard was estimated using the Nelson-Aalen estimator. The slopes of the curves equalize the hazard ratios: a slope of 1.00 corresponds to a hazard ratio of 1.00, whereas a slope >1 implies that colectomy patients have a higher risk of type 2 diabetes compared with non-colectomy patients. The linearity attests to the fulfillment of the assumption of proportional hazards in the Cox regression models.

DOI: https://doi.org/10.7554/eLife.37420.005

## Discussion

In the present nationwide study, we observed an increased risk of diabetes in colectomy patients. The risk was highest among individuals who had the left part of the colon removed, whereas resection of the rectum was not associated with any change in risk of diabetes. Moreover, we observed an increased risk of diabetes irrespective of whether the patients had co-occurring colorectal cancer or other underlying diseases.

A major strength of the present work is its nationwide nature allowing inclusion of many thousands of patients who have had colectomies and a large number of non-colectomy patients leading to relatively narrow confidence intervals of the hazard ratios observed. Moreover, the data allowed by study design to control for known confounding by sex and age as well as the possible confounding factors implicit in the time period and factors influencing the decision-making to refer to and conduct surgery in general. However, various limitations of our study must be considered in the interpretation of the results as indicating a causal relation of colectomy to later diabetes risk. We did not have information on several other potential confounders, precluding adjustment for them. This implies the possibility that the reported estimates may be positively biased. Colorectal cancer is

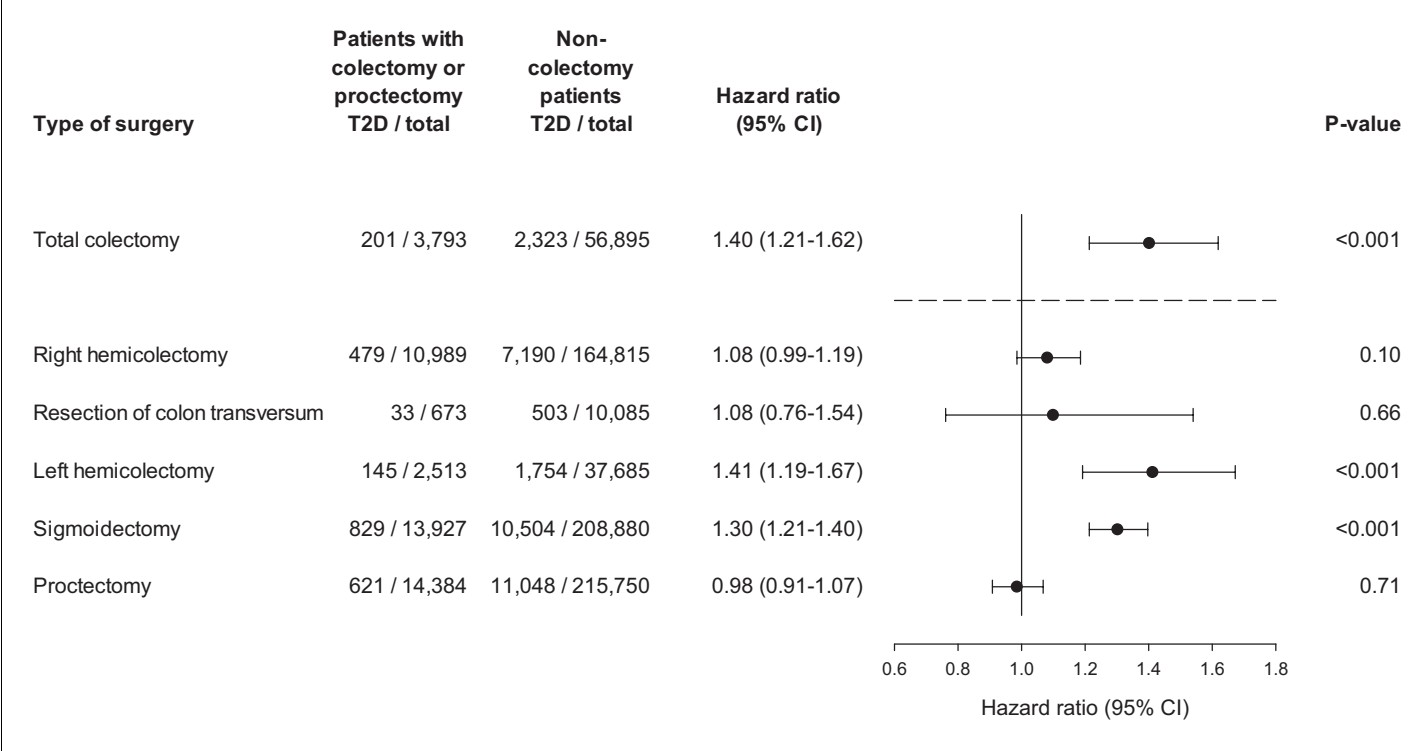

**Figure 3.** Hazard ratio of clinically recorded type 2 diabetes after colectomy. Hazard ratios are presented for total colectomy, right hemicolectomy, resection of colon transversum, left hemicolectomy, sigmoidectomy, and proctectomy. Total colectomy includes colectomy and proctocolectomy. Hazard ratios are adjusted for age, sex, and year of surgery. T2D: clinically recorded type 2 diabetes.
DOI: https://doi.org/10.7554/eLife.37420.006

**Table 2.** Hazard ratio of clinically recorded type 2 diabetes after total colectomy, left hemicolctomy, and sigmoidectomy for patients with co-occurring colon cancer and patients with other colorectal diseases.

|  | Patients with colectomy (T2D/total) | Non-colectomy patients (T2D/total) | Hazard ratio (95% CI) | P-value |
|---|---|---|---|---|
| **Total colectomy** | 201/3793 | 2323/56,895 | 1.40 (1.21–1.62) | <0.001 |
| Colorectal cancer | 56/993 | 620/14,895 | 1.61 (1.22–2.11) | <0.001 |
| Other colorectal diseases | 145/2800 | 1703/42,000 | 1.34 (1.13–1.59) | <0.001 |
| **Left hemicolectomy** | 145/2513 | 1754/37,685 | 1.41 (1.19–1.67) | <0.001 |
| Colorectal cancer | 93/1804 | 1271/27,050 | 1.24 (1.01–1.53) | 0.04 |
| Other colorectal diseases | 52/709 | 483/10,635 | 1.88 (1.41–2.50) | <0.001 |
| **Sigmoidectomy** | 829/13,927 | 10,504/208,880 | 1.30 (1.21–1.40) | <0.001 |
| Colorectal cancer | 334/6903 | 5071/103,520 | 1.11 (0.997–1.24) | 0.06 |
| Other colorectal diseases | 495/7024 | 5433/105,360 | 1.47 (1.34–1.62) | <0.001 |

Total colectomy includes colectomy and proctocolectomy. T2D: clinically recorded type 2 diabetes.
DOI: https://doi.org/10.7554/eLife.37420.007

associated with adiposity (*Ning et al., 2010*), which is also a major risk factor for diabetes, wherefore adiposity may confound our results. However, based on previously published data (*Berentzen et al., 2011*), it can be calculated that if differences in BMI were to fully account for our observed HR of 1.4 of diabetes, a BMI difference of 3 units (equaling a difference of 9 kg for individuals 175 cm tall) would be necessary. Another important potential confounder that we did not account for is medication. However, the direction of medication effects on risk of diabetes is likely ambiguous. For example, whereas corticosteroids used for treatment of inflammatory bowel disease have well-documented side-effects resulting in impaired glucose regulation, biologicals such as tumor necrosis factor-α antagonists may alleviate inflammation and therefore potentially improve glucose regulation (*Antohe et al., 2012*; *Parmentier-Decrucq et al., 2009*; *Solomon et al., 2011*; *Tam et al., 2007*). As the effects of medication on glucose regulation are ambiguous, and since we observed an enhanced risk of diabetes among patients with colorectal cancer and other colorectal diseases, it seems unlikely that our findings are due to differences in medication. Also, confounding by indication, that is, the indication for the colectomy — in this case the underlying disease — may bias our results. Hence, the underlying disease may, either through its effect on the normal colonic function or via other pathways, directly increase the risk of diabetes. Likely, the fact that we achieved similar results when we started the follow-up time 1000 days after the surgery in our main analyses or 1500 days after surgery in the supplemental analyses may speak against such confounding. Last, differences in the life style, for example in physical activity, smoking, dietary habits, including alcohol and coffee drinking, of the colectomy versus the non-colectomy groups may have played a role as confounders.

Another potential limitation of our study is that the outcome was based on data from the Danish National Patient Register, which includes data only on hospital inpatient and outpatient contacts. This implies that patients with undiagnosed diabetes and with diabetes diagnosed outside hospitals, but not recorded later in the hospitals, were missed, and that the observed incidence is an underestimation. On the other hand, it is also possible that the recorded diagnosis of diabetes is wrong. However, as we defined the non-colectomy group as individuals who had undergone various other types of surgery, we presume that the likelihoods of being referred to a hospital or an outpatient clinic, and thus the likelihoods of being registered with a clinical diagnosis of diabetes, are the same in the colectomy and non-colectomy patients. Further, the outcome diabetes based on the diagnosis code E14 (unspecified diabetes) may be subject to misclassification regarding the type of diabetes. The data available does not allow validation of the recorded diagnosis, but to avoid misclassification of type 2 diabetes as type 1 diabetes, we restricted the study population to individuals above 30 years, since type 1 diabetes is much more frequently diagnosed before than after that age. Also, type 1 diabetes is infrequent compared to type 2 diabetes, reducing the risk of significant misclassification in this regard (*Diabetesforeningen, 2018* Facts about diabetes in Denmark https://diabetes.dk/diabetesforeningen/in-english/facts-about-diabetes-in-denmark.aspx).

If the observed association between colectomy and excess incidence of diabetes is not simply explained by chance, bias, or confounding, it may represent a causal effect of colectomy on the diabetes risk. Indeed, our prospective design, including a 1000 days delay after surgery, supports causality since the outcome occurred after the exposure and the risk remained equally elevated throughout the follow-up periods. The finding of increased risk following only colectomy including the left part of the colon, but not the remainder part or the rectum, suggests a specific causal effect of that part rather than a general effect of surgery on the colon, which would more likely reflect confounding. Obviously, experimental proof of a direct causal effect of colectomy on diabetes risk in humans is impossible, and epidemiological substitutes, such as instrumental variable analyses, for example by Mendelian randomization, may not be feasible. However, assuming causality may inspire investigations of the possible biological mechanisms that may have induced the association between colectomy and diabetes risk.

Thus, it may be speculated that our findings are explained by removal of GLP-1 producing cells or alterations of the colonic gut microbiota composition and function. Recent studies suggest that modifications in GLP-1 release are directly involved in both development of diabetes (*Færch et al., 2015*) and diabetes remission after gastric bypass (*Holst and Madsbad, 2016*). However, the physiological role of GLP-1 secreted from the colon is unclear. Interestingly, the L-cell density increases from the proximal to the distal colon (*Eissele et al., 1992*; *Gunawardene et al., 2011*), and accordingly, we observed the highest risk among individuals who had the left part of the colon removed.

The bacterial load also increases along the length of the colon (*Donaldson et al., 2016*), and colectomies involving the left part of the colon therefore likely lead to removal of a larger part of the colonic microbiota than colectomies involving the right part of the colon. Notably however, it has been shown that ileocolonic resection leads to an altered colonization of the neo-terminal ileum (*GETAID et al., 2016*) suggesting that removal of the colonic gut microbiota may be somewhat compensated for by over-colonization of the remaining parts of the gastrointestinal tract. Such bacterial overgrowth may well change the microbiota-mediated conversion of primary bile acids to secondary bile acids, which through their hormone-like actions impact glucose metabolism (*Ridlon et al., 2014*). In our study, proctectomy was, with quite narrow upper confidence limits, not associated with an increased risk of diabetes. This may be explained by the fact that feces, and thus bacteria and other stimuli of L-cells, is only present in the rectum during the short time periods preceding defecation, thus hindering continuous signaling between the lumen of the rectum and extra-intestinal tissues. Last, the removal of the entire colon or a part of it might influence the exchanges between the gut epithelium and ingested food and fluids, especially due to an altered bacterial degradation of otherwise indigestible fibers to short-chain fatty acids, which could in turn have an impact upon diabetes development (*Allin et al., 2015*).

Among the previous studies that have examined glucose regulation in colectomy patients (*Besterman et al., 1982*; *Nauck et al., 1996*; *Palnaes Hansen et al., 1997*), the largest included 10 patients with an ileo-anal reservoir and 11 patients with an ileostomy who underwent an oral glucose tolerance test (*Palnaes Hansen et al., 1997*). Compared with 10 healthy individuals, colectomy individuals had higher peak levels of plasma glucose and higher peak levels and area under the curve of plasma insulin suggesting impaired glucose tolerance and hyperinsulinemia. However, the healthy controls had lower body weight compared with the patients, and no major differences in plasma GLP-1 levels were observed (*Palnaes Hansen et al., 1997*). The prior evidence from these studies for an association between colectomy and impaired glucose regulation is thus conflicting. In a previous study also based on the Danish National Patient Register, we found no corresponding difference in cardiovascular disease risk between individuals who had undergone a total colectomy and individuals who had undergone other types of surgeries leaving the gastrointestinal tract intact (*Jensen et al., 2015*). While we would expect that an increased risk of diabetes would associate with subsequent increased risk of cardiovascular diseases, this discrepancy may be explained by the limited follow-up time, not allowing presence of diabetes yet to be manifested in increased cardiovascular risk.

In conclusion, we observed an increased risk of clinically recorded type 2 diabetes among patients who had undergone total and partial colectomy with the risk being elevated only among individuals who had the left part of the colon removed. The increased risk of diabetes was observed in patients with colorectal cancer as well as in patients with other colorectal diseases. As our findings are based solely on register data, clinical-physiological studies are warranted to establish whether colectomy per se is associated with metabolic alterations indicating an increased risk of type 2 diabetes, and to understand the role of the colon in glucose homeostasis. Such studies could include comparisons of plasma glucose, insulin, and incretin hormone levels during a meal tolerance test before and after different types of colectomy with careful control for likely confounding factors such as those discussed above.

## Materials and methods

### Study population and design

The study is based on data from the Danish National Patient Register, which covers hospital inpatient and outpatient contacts for Danish patients from 1994 and onwards (*Lynge et al., 2011*). Based on the Nordic Classification of Surgical Procedures (NCSP, 1996 – 2015) and the Danish Surgical Procedure and Treatment Classification version 3 (DOTC, 1994 – 1995), we identified patients who had undergone total colectomy (with and without resection of rectum), partial colectomy, or resection of the rectum (proctectomy). We grouped partial colectomies into right hemicolectomy, resection of the transverse colon, left hemicolectomy, and sigmoidectomy (*Supplementary file 3*). We considered patients who had total colectomy (without resection of rectum) followed by resection of rectum

as having a proctocolectomy at the date of rectum resection. For patients who had partial and later total colectomy, we disregarded the preceding partial removals.

The selection of non-colectomy patients from the register was carried out among patients who had undergone other types of surgery not involving the gastrointestinal tract. Thus, the non-colectomy patients were selected from three groups of patients, and we selected five non-colectomy patients per colectomy patient from each group: (1) patients who had undergone orthopedic surgery, (2) patients who had undergone abdominal surgery leaving the gastrointestinal tract intact, and (3) patients who had undergone other surgery, unrelated to the gastrointestinal tract (*Supplementary file 4*). Non-colectomy patients were matched to colectomy patients by sex, year of birth (±1 year), and date of surgery (±1 year). If there were more than five patients available within each of the three surgical groups, we randomly selected five patients. The same series of matching patients may have been used for selection or non-colectomy patients for more than one colectomy patient implying that the individual non-colectomy patient may have been selected more than once (random sampling with replacement).

The outcome, clinically recorded type 2 diabetes, was defined as the International Classification of Diseases, 10th Edition (ICD-10) codes E11 and E14. Dates of deaths were obtained from the Danish Civil Registration System. Individuals who had a diagnosis of diabetes recorded before surgery were excluded.

The project has been reported to the Danish Data Protection Agency (ref 2015-54-0939) and the data extraction has been approved and delivered by Statens Serum Institut (ref FSEID-00001627). Informed consent and assessment of the proposal in scientific ethical committees are not required for registry-based research in Denmark.

## Statistical analysis

We analyzed the data using the 'surv package' in R version 3.2.2, and two-sided p-values<0.05 were considered statistically significant. The cumulative hazard of diabetes was estimated using the Nelson-Aalen estimator. Cox proportional hazards regression models were used to estimate hazard ratios (HRs) of diabetes. A linear relationship between the time-corresponding Nelson-Aalen estimators indicates fulfillment of the assumption of proportional hazards. We based our main analyses on those who were alive and who did not have a diagnosis of diabetes recorded during the first 1000 days after the surgery, for two reasons. First, as expected, colectomy patients had an increased hazard rate of death after surgery, which may reflect postsurgical morbidity possibly influencing diabetes risk for other reasons than effects of colectomy as such; the mortality among patients with total colectomy decreased during this period and stabilized from about 1000 days after surgery (*Supplementary file 5*). Second, diabetes may be associated with the diseases leading to colectomy, and may still be undiagnosed or not recorded at the time of surgery, but possibly so during the first years following surgery. The follow-up time for each participant ended at occurrence of diabetes, death, or at the end of the study, December 18th 2015, whichever came first, and the underlying time scale was time from 1000 days after surgery. None of the participants were lost during follow-up. In supplementary analyses, we examined the risk of diabetes when follow-up time was started at the date of surgery, 500 days after surgery, and 1500 days after surgery. To optimize the efficiency and minimize bias (*Sjölander and Greenland, 2013*) of the comparisons of the colectomy and non-colectomy patients, the Cox regression models included adjustment for the matching variables, sex, age and year of surgery. Separate models were fitted for total colectomy and for each type of partial colectomy and proctectomy and their respective non-colectomy patients. To elucidate whether the underlying disease, for which the colectomy was performed, influenced the subsequent risk of diabetes, the study population was stratified into two groups: a group including patients with co-occurrence of colorectal cancer (ICD-10 codes C18-C20) at the date of colectomy and a group including the remaining patients (*Supplementary file 1*).

## Acknowledgements

We thank Teresa Ajslev, Ph.D. for constructive comments during the preparation of the present study.

## Additional information

### Funding

| Funder | Grant reference number | Author |
|---|---|---|
| Horizon 2020 | MedBioinformatics - Research and Innovation Programme (grant agreement no. 634143) | Anders B Jensen |
| The Novo Nordisk Foundation | NNF16OC0022586 | Kristine H Allin |
| The Danish Diabetes Association | | Kristine H Allin |
| The Novo Nordisk Foundation | NNF17OC0027594 | Anders B Jensen Søren Brunak |
| The Novo Nordisk Foundation | NNF14CC0001 | Anders B Jensen Søren Brunak |

The funders had no role in study design, data collection and interpretation, or the decision to submit the work for publication.

### Author contributions

Anders B Jensen, Data curation, Software, Formal analysis, Funding acquisition, Visualization, Methodology, Writing—original draft, Writing—review and editing; Thorkild IA Sørensen, Conceptualization, Formal analysis, Supervision, Visualization, Methodology, Project administration, Writing—review and editing; Oluf Pedersen, Writing—review and editing; Tine Jess, Supervision, Writing—review and editing; Søren Brunak, Conceptualization, Resources, Software, Supervision, Funding acquisition, Methodology, Project administration, Writing—review and editing; Kristine H Allin, Funding acquisition, Visualization, Writing—original draft, Writing—review and editing

### Author ORCIDs

Thorkild IA Sørensen (iD) http://orcid.org/0000-0003-4821-430X
Kristine H Allin (iD) http://orcid.org/0000-0002-6880-5759

### Ethics

Human subjects: The project has been reported to the Danish Data Protection Agency (ref 2015-54-0939) and the data extraction has been approved and delivered by Statens Serum Institut (ref FSEID-00001627). Informed consent and assessment of the proposal in scientific ethical committees are not required for registry-based research in Denmark.

### Decision letter and Author response

Decision letter https://doi.org/10.7554/eLife.37420.015
Author response https://doi.org/10.7554/eLife.37420.016

## Additional files

### Supplementary files

• Supplementary file 1. Diagnoses for patients with colectomy who did not have colorectal cancer.
DOI: https://doi.org/10.7554/eLife.37420.008

• Supplementary file 2. Hazard ratio of clinically recorded type 2 diabetes after colectomy according to start of follow-up time.
DOI: https://doi.org/10.7554/eLife.37420.009

• Supplementary file 3. Procedure codes for colectomies and proctectomies used to identify patients in the Danish National Patient Register.
DOI: https://doi.org/10.7554/eLife.37420.010

• Supplementary file 4. Procedure codes used to identify non-colectomy patients in the Danish National Patient Register.
DOI: https://doi.org/10.7554/eLife.37420.011

• Supplementary file 5. Cumulative hazards plots of survival for patients with total colectomy, right hemicolectomy, resection of colon transversum, left hemicolctomy, sigmoidectomy, proctectomy, and non-colectomy patients.
DOI: https://doi.org/10.7554/eLife.37420.012

• Transparent reporting form
DOI: https://doi.org/10.7554/eLife.37420.013

### Data availability

The study is based on data from the Danish National Patient Register [https://sundhedsdatastyrelsen.dk]. Procedures codes from the Danish National Patient Register are provided in Supplementary files 3 and 4. The Danish National Patient Register is protected by the Danish Act on Processing of Personal Data and can be accessed through application to and approval from the Danish Data Protection Agency and the Danish Health Data Authority [https://sundhedsdatastyrelsen.dk/da/forskerservice/ansog-om-data] where the purpose and the feasibility of the intended analysis should be accounted for. Informed consent and assessment of the proposal in scientific ethical committees are not required for registry-based research in Denmark.

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
