## [Decision Letter]

Thank you for submitting your article "Risk of type 2 diabetes after colectomy" for consideration by *eLife*. Your article has been reviewed by three peer reviewers, one of whom is a member of our Board of Reviewing Editors, and the evaluation has been overseen by Prabhat Jha as the Senior Editor. The reviewers have opted to remain anonymous.

The reviewers have discussed the reviews with one another and the Reviewing Editor has drafted this decision to help you prepare a revised submission.

The authors present data from a record linkage study in Denmark which has been used to investigate the association between having different types of partial or complete colectomy and future diagnosis of diabetes. The purpose behind such a study is the investigation of the role of the colon in the aetiology of diabetes.

Summary:

The observation made in this paper of an association between colectomy and incident diabetes is potentially important but there are questions about possible confounding that need to be addressed in greater depth and also needs to be mentioned in the Abstract. The authors should follow the suggestion to restructure the Discussion to focus first on the question of whether or not there is a real association before proceeding to discuss whether or not it is likely to be causal and what biological mechanisms may underlie it.

As with all such observational studies the first question is whether there is an association between colectomy and diabetes incidence and secondly whether that association is explained by chance, bias or confounding. The authors have done a good job in largely ruling out the role of chance since the study has been conducted in the context of a national health data registry thus allowing them to include many thousands of patients who have had colectomies and a large number of control patients. The confidence intervals of the hazard ratios observed are generally relatively narrow. However, the problems of confounding are much more difficult to deal with and the paper needs to focus more on how this may affect the results.

Essential revisions:

1) The researchers have included all patients who have had partial or total colectomy within the national system of Denmark within a specific time period, thus the possibility of selection bias is limited. The national register includes data from hospital inpatient and outpatient contacts and therefore there is limited possibility of information bias either. The outcome is clinically diagnosed type 2 diabetes according to the ICD-10 classification given in the register. We think it would be worth describing this throughout as "clinically diagnosed type 2 diabetes" instead of simply saying that the research team as studied the incidence of type 2 diabetes (T2D). The addition of "clinically diagnosed" makes two points; firstly that there is a distinction between all diabetes and that which is diagnosed. This is important because any association with diagnosed T2D could be a true association with diabetes or alternatively an association with an increased likelihood of it being diagnosed. Secondly it makes the point that the diagnosis of T2D as opposed to any other form of diabetes is clinical and thus subject to misclassification. The Discussion section of the paper describes the potential for the case ascertainment process to miss cases, but it doesn't describe the potential for misclassification.

2) The major limitation of the analysis, as acknowledged by the authors, is the lack of information on potential confounding variables. However, given these are individuals within the hospital system we were surprised that basic data such as BMI were not available. Can the authors confirm such data are not captured by the Danish hospital system?

3) The non-colectomy patients are highly selected using individual matching on age, sex and date of surgery, resulting in identical means, SDs and% s in those with and without colectomy (Table 1), so it is unclear what the point was of also adjusting for these variables in the Cox regression models.

4) Of greater concern is the fact that no other potential confounders were included in the model. The authors highlight this as an issue in the fourth paragraph of the Discussion and discuss why the result may not be explained by adiposity or medication; however, even if there is a true association, the inability to adjust for multiple potential confounders (not just BMI and medication, but also activity, diet and smoking) means that the reported estimates are likely to be positively biased, so the statement that the work "enables a precise estimation of the association" is highly questionable. The lack of data on and adjustment for confounders is a major limitation which should also be stated in the Abstract.

5) In the Discussion, it is noted: "However, we observed a similar risk of type 2 diabetes in patients with colorectal cancer or other colorectal diseases suggesting that adiposity does not account for our results." However, adiposity may also be a risk factor for the other colorectal diseases (whatever they might be). In any case, the authors really cannot get around the lack of such data and the interpretation of the results should be conducted with this in mind.

6) Figure 2 claims to plot cumulative hazard for colectomy vs. non-colectomy patients, but cumulative hazard is a function of time, so it is not clear what is being presented here, or whether this type of plot is actually useful. It would be preferable to see Kaplan-Meier plots of cumulative survival (or failure) over time for the colectomy and non-colectomy groups. The statement “Compared with the control patients, the cumulative hazard of type 2 diabetes was greater among patients who had undergone total colectomy (Figure 2)” also needs to be clarified.

7) Having considered whether the association is true and not a reflection of chance, bias and confounding, the next question is whether it is likely to be causal. Of the criteria used to investigate the likelihood of causality, biological plausibility is one of the weakest but within the Discussion section of this paper is given rather too much prominence and is considered almost at the beginning of the discussion and is quite lengthy. A restructuring of the Discussion might be considered so that it is clear to begin with that the observation is an association between colectomy and incidence of clinically diagnosed T2D. Alternative explanations for the observed association should then be considered before the issues of possible causality are discussed, of which biological plausibility is only one.

8) Even if it is cited in the Introduction (first paragraph), the role of the gut in the absorption of nutrients is not considered a possible factor in influencing the development of diabetes. However, the removal of the entire colon or a part of it might influence the exchanges between the gut epithelium and ingested food and fluids, and hence modify the absorption of nutrients which could in turn impact upon diabetes development. This should be indicated as a possible explanation for why colectomy might raise diabetes risk.

9) The final section of the Discussion mentions the need for "clinical-physiological studies" to establish whether colectomy per se is associated with an increased risk of diabetes. It is not clear what the authors mean by this and this should be clarified. They also state that an implication of this paper is that clinicians should be aware of this risk to improve the long-term health of this patient group. Given that the risk increase is relatively small and the authors have been unable to deal with confounding by a whole series of individual-level confounding factors, it does seem a bit of a leap to suggest that a major implication of this study is clinical. It is more likely that the implications lie in understanding the role of the colon in glucose homeostasis.

---

## [Author Response]

Essential revisions:1) The researchers have included all patients who have had partial or total colectomy within the national system of Denmark within a specific time period, thus the possibility of selection bias is limited. The national register includes data from hospital inpatient and outpatient contacts and therefore there is limited possibility of information bias either. The outcome is clinically diagnosed type 2 diabetes according to the ICD-10 classification given in the register. We think it would be worth describing this throughout as "clinically diagnosed type 2 diabetes" instead of simply saying that the research team as studied the incidence of type 2 diabetes (T2D). The addition of "clinically diagnosed" makes two points; firstly that there is a distinction between all diabetes and that which is diagnosed. This is important because any association with diagnosed T2D could be a true association with diabetes or alternatively an association with an increased likelihood of it being diagnosed. Secondly it makes the point that the diagnosis of T2D as opposed to any other form of diabetes is clinical and thus subject to misclassification. The Discussion section of the paper describes the potential for the case ascertainment process to miss cases, but it doesn't describe the potential for misclassification.

We accept the arguments for making this aspect more explicit. However, we suggest that it would be even more precise to say, ‘clinically recorded type 2 diabetes’. In the present nationwide study population, the diagnosis of diabetes will always be the result of a clinical procedure carried out at the hospital contact, where it is recorded or earlier, e.g. by the general practitioners. We use the information about type 2 diabetes when it is recorded for the first time in the patient register, i.e. at a hospital contact. Therefore, we have specified that the study is based on ‘clinically recorded type 2 diabetes’ in the title (which has consequently been slightly modified), the Abstract, the end of the Introduction, the beginning of the Results section, the Materials and methods section, and in the headings of the tables and the figure legends, but we suggest that it makes the text rather heavy-going by using the full phrase throughout the manuscript.

We have added a description of the possible misclassification regarding the type of diabetes in the Discussion section:

“Further, the outcome diabetes based on the diagnosis code E14 (unspecified diabetes) may be subject to misclassification regarding the type of diabetes. […] Also, type 1 diabetes is infrequent compared to type 2 diabetes, reducing the risk of significant misclassification in this regard ("Diabetesforeningen. Facts about diabetes in Denmark (https://diabetes.dk/diabetesforeningen/in-english/facts-about-diabetes-in-denmark.aspx)").”

2) The major limitation of the analysis, as acknowledged by the authors, is the lack of information on potential confounding variables. However, given these are individuals within the hospital system we were surprised that basic data such as BMI were not available. Can the authors confirm such data are not captured by the Danish hospital system?

Unfortunately, we do not have access to individual hospital records, where such information may be included. We can confirm that data on BMI is not captured in the Danish National Patient Register. The register contains information such as hospital departments, diagnoses and operation codes, and date for hospital arrival and departure, outpatients contacts, and surgical procedures. However, no information on anthropometrics is registered.

3) The non-colectomy patients are highly selected using individual matching on age, sex and date of surgery, resulting in identical means, SDs and% s in those with and without colectomy (Table 1), so it is unclear what the point was of also adjusting for these variables in the Cox regression models.

The sampling procedure implies that the most efficient and unbiased comparison is based on each colectomy patient with the matched non-colectomy patients. The non-colectomy patients matched to other colectomy patients different in sex, age and date of surgery (of potential great influence on risk of diabetes) may not be valid for that comparison. To maintain these strengths of the matched comparisons, we added the matching variables to the regression models. Accordingly, in a methods paper from Sjölander and Greenland (2013) conclude that ‘ignoring the matching variables in a cohort study can leave bias if there are additional confounders, even with adjustment for the additional confounders. This bias can be avoided by adjusting for the matching variables.’

We have modified the text in the Materials and methods section (including a reference to the above-mentioned paper) accordingly:

“To optimize the efficiency and minimize bias (Sjolander & Greenland, 2013) of the comparisons of the colectomy and non-colectomy patients, the Cox regression models included adjustment for the matching variables, sex, age and year of surgery.”

4) Of greater concern is the fact that no other potential confounders were included in the model. The authors highlight this as an issue in the fourth paragraph of the Discussion and discuss why the result may not be explained by adiposity or medication; however, even if there is a true association, the inability to adjust for multiple potential confounders (not just BMI and medication, but also activity, diet and smoking) means that the reported estimates are likely to be positively biased, so the statement that the work "enables a precise estimation of the association" is highly questionable. The lack of data on and adjustment for confounders is a major limitation which should also be stated in the Abstract.

We agree that there may be a risk of uncontrolled confounding that leads to positively biased estimates, and we also agree that there may be other known, as those mentioned, or even unknown confounders beyond those focused upon in our Discussion.

With the statement about ‘precise estimation’, we refer to the narrowness of the confidence intervals. However, we acknowledge that it may be mistaken for ‘accuracy’ (or ‘validity’), which may well be affected by the possible biases and confounding. Accordingly, we have deleted the statement, and instead, based on the general comment above, we say:

“A major strength of the present work is its nationwide nature allowing inclusion of many thousands of patients who have had colectomies and a large number of matched non-colectomy patients leading to relatively narrow confidence intervals of the hazard ratios observed.”

Additionally, we have added this sentence to the Abstract:

“Although we were not able to adjust for several potential confounders, our findings …”.

5) In the Discussion, it is noted: "However, we observed a similar risk of type 2 diabetes in patients with colorectal cancer or other colorectal diseases suggesting that adiposity does not account for our results." However, adiposity may also be a risk factor for the other colorectal diseases (whatever they might be). In any case, the authors really cannot get around the lack of such data and the interpretation of the results should be conducted with this in mind.

We acknowledge this argument and accordingly, we have deleted this sentence from the Discussion.

6) Figure 2 claims to plot cumulative hazard for colectomy vs. non-colectomy patients, but cumulative hazard is a function of time, so it is not clear what is being presented here, or whether this type of plot is actually useful. It would be preferable to see Kaplan-Meier plots of cumulative survival (or failure) over time for the colectomy and non-colectomy groups. The statement “Compared with the control patients, the cumulative hazard of type 2 diabetes was greater among patients who had undergone total colectomy (Figure 2)” also needs to be clarified.

Kaplan-Meier plots of cumulative survival (or failure) probabilities would not be interpretable in the type of data analyzed here. There are two options for making such plots, both of which will be based on invalid assumptions. If patients who die during follow-up without having had the diagnosis of diabetes recorded are censored in the analysis, then the estimated probabilities in the Kaplan-Meier plots will assume that the patients who die are still alive with the same diabetes risk as before, which makes no sense. If a combined end-point of diabetes or death is used instead, the comparisons between the colectomy and non-colectomy patients will be biased because of different mortality in these groups, especially for the patients after total colectomy as shown in Supplementary file 5. These problems of using conditional cumulative probabilities are avoided by estimating Nelson-Aalen cumulative hazards over time as we have done it in Supplementary file 5. One option to show the time-dependency of the emergence type 2 diabetes (as recorded for the first time) could be to show such graphs for all types of colectomy and their respective non-colectomy groups. However, we think such graphs will be overwhelming and difficult to read, and they will add little to answering the key question about the difference in risk (hazard) over time across the colectomy and non-colectomy groups. By plotting the time-corresponding cumulative hazard (double Nelson-Aalen plot) as done in Figure 2, we get an optimal representation of the hazard ratios over time, because the slope of the curves equalizes the hazard ratios (a slope of 1.00 is a hazard ratio of 1.00). At the same token, these plots show that the fundamental assumption of the Cox regression analysis of proportional hazards over time is fulfilled by the curves being approximately straight lines. Therefore, we would suggest that we maintain the presentations as done in Figure 2. To secure the correct interpretation by the readers, we have added this text:

– Methods section: “The cumulative hazard of diabetes was estimated using the Nelson-Aale estimator. […] A linear relationship between the time-corresponding Nelson-Aalen estimators indicates fulfillment of the assumption of proportional hazards.”

– Results section: “Figure 2 shows the time-corresponding cumulative hazard (double Nelson-Aalen plot) of diabetes for colectomy patients vs. non-colectomy patients. The slopes of the curves equalize the hazard ratios (a slope of 1.00 corresponds to a HR of 1.00).”

– Figure 2 legend: “The cumulative hazard was estimated using the Nelson-Aalen estimator. […] The linearity attests to the fulfillment of the assumption of proportional hazards in the Cox regression models.”

Furthermore, we have clarified the statement: “The cumulative hazard of diabetes was greater among patients who had undergone total colectomy compared with non-colectomy patients.”

7) Having considered whether the association is true and not a reflection of chance, bias and confounding, the next question is whether it is likely to be causal. Of the criteria used to investigate the likelihood of causality, biological plausibility is one of the weakest but within the Discussion section of this paper is given rather too much prominence and is considered almost at the beginning of the discussion and is quite lengthy. A restructuring of the Discussion might be considered so that it is clear to begin with that the observation is an association between colectomy and incidence of clinically diagnosed T2D. Alternative explanations for the observed association should then be considered before the issues of possible causality are discussed, of which biological plausibility is only one.

Whereas we of course agree that biological plausibility is just one, and usually also a weak, indicator of causality, we do suggest that it is the possible biological explanations of the observations that make them particularly interesting from a biomedical point of view. However, we have complied with the request by restructuring the discussion, which now underlines the importance of confounding and bias first, followed by a discussion of possible causality and ending with a paragraph on biological plausibility, which has been shortened.

8) Even if it is cited in the Introduction (first paragraph), the role of the gut in the absorption of nutrients is not considered a possible factor in influencing the development of diabetes. However, the removal of the entire colon or a part of it might influence the exchanges between the gut epithelium and ingested food and fluids, and hence modify the absorption of nutrients which could in turn impact upon diabetes development. This should be indicated as a possible explanation for why colectomy might raise diabetes risk.

We agree that especially removal of the entire colon and perhaps the right part may affect the absorption, also because of compensatory mucosal adaptations in the small intestine. We find it less likely that isolated removal of the left colon or the sigmoid colon would have such general effects on nutrient absorption, but there may be effects on specific nutrients, e.g. short-chain fatty acids, produced in the colon by bacterial degradation of the otherwise indigestible fibers. We have added this sentence to the Discussion:

“Last, the removal of the entire colon or a part of it might influence the exchanges between the gut epithelium and ingested food and fluids, especially due to an altered bacterial degradation of otherwise indigestible fibers to short-chain fatty acids, which could in turn have an impact upon diabetes development (Allin et al., 2015).”

9) The final section of the Discussion mentions the need for "clinical-physiological studies" to establish whether colectomy per se is associated with an increased risk of diabetes. It is not clear what the authors mean by this and this should be clarified. They also state that an implication of this paper is that clinicians should be aware of this risk to improve the long-term health of this patient group. Given that the risk increase is relatively small and the authors have been unable to deal with confounding by a whole series of individual-level confounding factors, it does seem a bit of a leap to suggest that a major implication of this study is clinical. It is more likely that the implications lie in understanding the role of the colon in glucose homeostasis.

We have added this sentence to the Discussion:

“Such studies could include comparisons of plasma glucose, insulin, and incretin hormone levels during a meal tolerance test before and after different types of colectomy with careful control for likely confounding factors such as those discussed above.”

We have deleted the last sentence in the Discussion about the relevance of the findings for the clinicians.